# Effect of Dipole Interactions on Blocking Temperature and Relaxation Dynamics of Superparamagnetic Iron-Oxide (Fe_3_O_4_) Nanoparticle Systems

**DOI:** 10.3390/ma16020496

**Published:** 2023-01-04

**Authors:** Md Ehsan Sadat, Sergey L. Bud’ko, Rodney C. Ewing, Hong Xu, Giovanni M. Pauletti, David B. Mast, Donglu Shi

**Affiliations:** 1Department of Physics, University of Cincinnati, Cincinnati, OH 45221, USA; 2Ames Laboratory, Department of Physics and Astronomy, Iowa State University, Ames, IA 50011, USA; 3Department of Geological Sciences, Stanford University, Stanford, CA 94305-2115, USA; 4Med-X Institute, Shanghai Jiao Tong University, Shanghai 200030, China; 5Department of Pharmaceutical and Administrative Sciences, St. Louis College of Pharmacy, University of Health Sciences & Pharmacy, St. Louis, MO 63110, USA; 6The Materials Science and Engineering Program, Department of Mechanical and Materials Engineering, College of Engineering and Applied Science, University of Cincinnati, Cincinnati, OH 45221, USA

**Keywords:** Fe_3_O_4_ nanoparticles, magnetic relaxation, gyromagnetic resonance, blocking temperature

## Abstract

The effects of dipole interactions on magnetic nanoparticle magnetization and relaxation dynamics were investigated using five nanoparticle (NP) systems with different surfactants, carrier liquids, size distributions, inter-particle spacing, and NP confinement. Dipole interactions were found to play a crucial role in modifying the blocking temperature behavior of the superparamagnetic nanoparticles, where stronger interactions were found to increase the blocking temperatures. Consequently, the blocking temperature of a densely packed nanoparticle system with stronger dipolar interactions was found to be substantially higher than those of the discrete nanoparticle systems. The frequencies of the dominant relaxation mechanisms were determined by magnetic susceptibility measurements in the frequency range of 100 Hz–7 GHz. The loss mechanisms were identified in terms of Brownian relaxation (1 kHz–10 kHz) and gyromagnetic resonance of Fe_3_O_4_ (~1.12 GHz). It was observed that the microwave absorption of the Fe_3_O_4_ nanoparticles depend on the local environment surrounding the NPs, as well as the long-range dipole–dipole interactions. These significant findings will be profoundly important in magnetic hyperthermia medical therapeutics and energy applications.

## 1. Introduction

Superparamagnetic Fe_3_O_4_ nanoparticles exhibit unique physical and structural properties that have been utilized for a variety of energy [1] and biomedical applications [2]. While transparent Fe_3_O_4_ thin films can spectral-selectively harvest solar light to generate sufficient photothermal heat for energy-efficient building skins [1], local heat can also be produced in solutions of the Fe_3_O_4_ nanoparticles by alternating (AC) magnetic fields for cancer therapy [2]. In solar harvesting, the Fe_3_O_4_ nanoparticle size, geometry, and anisotropy play critical roles in the optical absorption behaviors based on the localized surface plasmon resonance (LSPR) [1]. It has been shown that the cancer cells can be effectively killed when the local temperature is raised between 42 °C and 45 °C [2]. The mechanism of magnetic hyperthermia has been explained by the so-called Néel and Brownian relaxations [2]. However, the frequency response of the magnetic moments in the Fe_3_O_4_ magnetic nanoparticles (MNPs) to the electromagnetic (EM) fields needs to be investigated for the enhanced magnetic hyperthermia effects that depend on particle size, inter-particle spacing, and particle–particle interactions [2,3]. At very high frequencies (3 MHz and above), it is well known that living cells, which contain permanent dipole moments, can interact with the electrical component of the RF radiation, thus contributing to dielectric heating. For magnetic heating, energy absorption in the living cells is mainly due to magnetic coupling. The radio frequency in this range has been shown to be not only biologically benign, but also sufficient in penetrating tissues for treating inner lesions in the body [4].

The hyperthermia heating capabilities of Fe_3_O_4_ MNPs in AC fields can be significantly enhanced by tuning the resonance frequency of the Néel relaxation time in the range of 100 kHz–1 MHz, which is particularly effective for in vivo therapies [4]. In the absence of interactions, both the Néel relaxation time and gyromagnetic resonance frequency depend on the anisotropy of the individual MNPs [5]. In real MNP systems, the energy loss mechanisms may depend on particle–particle interactions and can be modified by surface functionalization or by the physical confinement of the nanoparticles. However, there have been few studies correlating the intrinsic magnetic anisotropy of the particles to the frequency-dependent electromagnetic properties [6]. Previous studies have shown that magnetic dipole interactions play an important role in determining the MNPs “effective” magnetic anisotropy ($K_{eff}$), which in turn affects the MNPs specific absorption rate (SAR) [7,8]. However, there is little evidence in the literature that correlates the dipole interactions and “effective” magnetic anisotropy to the frequency-dependent electromagnetic properties of MNPs and hyperthermia heating.

As has been well established, single-domain particles generate heat when exposed to a high-frequency magnetic field, through the oscillation of their magnetic moments [6,7,8]. However, energy dissipation can be attributed to both Brownian and Néel relaxation mechanisms. The former tends to dominate if the particles have more freedom to rotate and oscillate in the liquid. It has been well demonstrated that Brownian relaxation can be significantly reduced in frozen magnetic fluids [9]. By densely packing the nanoparticles in the spherical polystyrene matrices, similar behaviors can be observed due to difficult rotational movements of the particles. Under the same AC fields, the ferrofluid systems (i.e., the particles are dispersed in liquid) may exhibit significant heating, where the particles are free to rotate and move in the liquids, resulting in both Brownian and Néel relaxations. By comparing the frequency dependence of these two types of systems, one can distinguish between the contributions of the Brownian and Néel relaxation mechanisms. In most previous studies [10,11,12], researchers mainly focused on the frequency dependence of magnetic hyperthermia heating, but not on the correlations between inter-particle interactions and magnetic anisotropy. 

We systematically investigated magnetization, relaxation dynamics, and the effective magnetic anisotropy properties of different surface-modified Fe_3_O_4_ MNP systems. These magnetic systems were developed with different inter-particle spacing, particle confinement, and surface chemistry. Correlations were established between these MNP characteristics and effective magnetic anisotropy and relaxation time for a fundamental understanding of the heating mechanisms.

## 2. Experimental Details

The surfaces of the MNP samples were functionalized with different polymers, providing varied surface morphologies, chemical properties, particle spacing, and physical confinement. These systems are as follows: (1) commercial oil-based ferrofluids with discrete Fe_3_O_4_ MNPs (~10 nm diameter) dispersed in a hydrocarbon carrier (EMG 900 and EFH1 (FerroTec, Bedford, NH, USA)); (2) commercial water-based ferrofluid with discrete Fe_3_O_4_ MNPs (~10 nm diameter) coated with an anionic surfactant (EMG 700, FerroTec, USA) and cationic surfactant (EMG 605, FerroTec, Bedford, NH, USA); and (3) Silicon/Polystyrene/Fe_3_O_4_ (denoted as Si/PS/Fe_3_O_4_), a water-based nanocomposite composed of the Fe_3_O_4_ nanoparticles (~10 nm) embedded in the matrices of polystyrene spheres with an averaged particle diameter of ~100 nm [3]. The synthesis procedures can be found in several previously published papers [2,3]. Briefly, a chemical coprecipitation method [3] was employed to synthesize the Fe_3_O_4_ nanoparticles with an average particle size ∼10 nm. The nanoparticles were dispersed in situ in octane to form a ferrofluid. A miniemulsion was developed by adding the Fe_3_O_4_ ferrofluid to the aqueous solution with sodium dodecyl sulfate (SDS) as surfactant. This miniemulsion containing Fe_3_O_4_ aggregations was mixed with another miniemulsion made of the styrene monomer droplets. The mixture of both miniemulsion was reacted at 80 °C for 20 h to form the polystyrene nanospheres with Fe_3_O_4_ nanoparticles embedded in the spherical matrices. The nanosphere surfaces were encapsulated in silica by following a previously reported process [3] to complete the synthesis of Si/PS/Fe_3_O_4_. The concentrations of the MNP systems were varied by diluting the oil-based ferrofluids (EMG 900 and EFH1) with Isopar-M (viscosity η = 2.06 mPa-s, freezing point = 196 K), and by diluting the water-based ferrofluids (EMG 700 and EMG 605) with distilled water. 

The average core diameter and size distribution of the nanoparticles in the commercial ferrofluids were confirmed by transmission electron microscopy (TEM) (Figure 1a) (figure supplied by FerroTec, USA). Using the dynamic light scattering technique (DLS), the average hydrodynamic diameter and lognormal size distribution (measured by Zeta Sizer, Malvern Instrument) of the nanoparticle systems were determined to be 263.03 nm for EMG 700; 591.22 nm for EMG 605; 44.93 nm for EMG 900; and 40.92 nm for EFH1, with a standard deviation of 0.474 ± 0.013; 0.775 ± 0.030; 0.414 ± 0.013; and 0.0461 ± 0.012, respectively. Considerable agglomeration was observed for EMG 605, which resulted in a large hydrodynamic diameter. 

The TEM images and the hydrodynamic diameter of Si/PS/Fe_3_O_4_ have been reported in our previous studies [2,13,14]. The TEM images were taken using a JEOL 2010F for the characterization of the nanoparticles. Samples were developed by placing a drop of the MNP solution on a carbon coated copper grid and letting it dry at room temperature. The mean hydrodynamic diameter and size distribution of the MNPs dispersed in water were measured using a Zetasizer (Nano Series, Malvern Panalytical, Westborough, MA, USA). Figure 1b shows the transmission electron microscopy (TEM) image of the Si/PS/Fe_3_O_4_ sample. As can be seen in Figure 1b, the nanoparticles of Fe_3_O_4_ (darker spots) were densely packed in the spherical polystyrene matrices (lighter spheres), thus physically confined without the freedom to rotate, or oscillate under AC fields. This condition was significantly different from the ferrofluid systems (e.g., the EMG samples), where the nanoparticles of Fe_3_O_4_ were dispersed in liquid. When these samples were exposed to AC fields, their behaviors were found to be drastically different due to dipole–dipole interactions associated with inter-particle spacing and physical confinement. 

By comparing Figure 1a,b, one can see the distinctive characteristics of the Fe_3_O_4_ nanoparticles and the Si/PS/Fe_3_O_4_ nano spheres. For Si/PS/Fe_3_O_4_, the Fe_3_O_4_ nanoparticles were physically confined in the polystyrene spherical matrices. In this situation, the inter-particle separation was small, which was of the order of one particle diameter, resulting in strong dipole interactions. However, in ferrofluid, the Fe_3_O_4_ nanoparticles were free to move in the solution with much weaker inter-particle interactions. These two unique behaviors will result in significant differences in magnetic dipole interactions.

The zero-field cooled (ZFC) and field cooled (FC) DC magnetization data of the ferrofluid samples were taken using a superconducting quantum interference device (SQUID) magnetometer (MPMS 5). The frequency dependent complex susceptibility of the nanoparticles systems was measured over a frequency range of 100 Hz–10 MHz using the slit toroid technique described by Fannin et al. with the Solartron (Model 1260) Impedance/Gain Phase analyzer [15]. The high frequency complex permeability of the ferrofluids was measured using the short-open coaxial transmission line technique (SOCL), as reported by Bellizzi et al., using a custom built open and short circuit cavities with sample lengths of 2.1 mm and 4.2 mm [16].

## 3. Results and Discussion

Figure 2a shows the magnetic field dependence of the magnetization, *M(H)*, of two water-based samples, EMG 605 and EMG 700, at a concentration of 48 mg/mL. As can be seen from this figure, both ferrofluids exhibited very low retentivity and coercivity ($H_{c}$) with a small hysteresis area observed in the low-field portions of the data (inset of Figure 2a). The area of the hysteresis loop (minor loop) was calculated to be 74.63 Oe-emu/mL for EMG 700 and 49.16 Oe-emu/mL for EMG 605. It has been reported in previous works that for a given particle size, the hysteresis loss decreases with the increasing dipole interactions [11,17]. Consistent with these previous studies, the lower hysteresis loop area of EMG 605, compared with that of EMG 700, was associated with stronger inter-particle interactions due to severe aggregation in this sample. The field dependent magnetization behavior of Si/PS/Fe_3_O_4_ was reported in our previous study [2]. For 10 mg/mL Si/PS/Fe_3_O_4_, the hysteresis loop area was found to be even smaller, 7.74 Oe-emu/mL, due to densely packed Fe_3_O_4_ particles physically confined in the polystyrene matrices [2]. The magnetization *M(H)* behaviors of two Isopar-M-based ferrofluids were also investigated at a concentration of 48 mg/mL [Figure 2b]. For EMG 900 and EFH1, the minor hysteresis loop areas were determined to be 99.47 Oe-emu/mL and 129.07 Oe-emu/mL, respectively. The larger hysteresis loop of EFH1 was due to its larger distribution of particle sizes compared with EMG 900 (confirmed from zeta sizer measurements and manufacturer data). 

In the Stoner–Wohlfarth model, for monodispersed, single-domain particles, the hysteresis loss is predicted to be proportional to the product of retentivity and coercivity. However, the experimental investigations of Heider et al. [18] and Hergt et al. [19] on different diameter particles showed that the hysteresis loss was in fact proportional to the coercivity ($H_{c}$). A general expression for coercivity ($H_{c}$) as a function of particle diameter (*D*) was reported by Hergt et al. [19] and can be represented as follows:(1)$$H_{c}\left(D\right)=H_{M}{\left(\frac{D}{D_{1}}\right)}^{-0.6}\left[1-e^{{\left(-\frac{D}{D_{1}}\right)}^{5}}\right]$$
where $H_{M}$ and $D_{1}$ are the materials specific parameters. Hergt et al. plotted the coercivity as a function of the particle diameter (from the superparamagnetic to single domain to multidomain) for magnetite, with a typical value of $H_{M}$ = 32 kAm^−1^ and $D_{1}$= 15 nm. A value of coercivity ($H_{c}$) = ~4.2 kAm^−1^ can be determined from their data for a mean particle diameter of 10 nm, while we obtained a value of $H_{c}$ = 0.904 kAm^−1^ for the EMG 900 and EFH1 samples. This difference in coercivity between our samples and those reported in [19] can be ascribed to the dissimilar materials parameter used in Hergt’s model. However, it is obvious from Equation (1) that, for a larger distribution of the particle diameter, the hysteresis loss would be greater, as observed in our study. It should be noted that the coercive field is also temperature and frequency dependent. In this study, we focused on the size dependence of the coercive field by comparing different particles systems in terms of their size and inter-particle spacing variations.

Inter-particle interactions in magnetic nanoparticles have been extensively investigated [20,21]. Zero-field cooled (ZFC) and field cooled (FC) temperature dependent magnetization was used to investigate the inter-particle interactions of the samples with different carrier fluids, surfactants, and size distributions. For superparamagnetic systems, ZFC magnetization exhibits a maximum at the so-called blocking temperature, defined by: τB=KeffVkBln(τmτo), where *V* is the volume of the particle, $K_{eff}$ is the effective magnetic anisotropy, *k_B_* is the Boltzmann constant, and $\tau_{o}$ is the attempt time, typically in the order of $10^{-9}$ to $10^{-10}$ s. This blocking temperature is a measure of the strength of the collective magnetic ordering and relaxation mechanisms of MNPs. The easy axis of the MNPs is oriented in a random direction for assembly of the nanoparticles in a thermal equilibrium. As the temperature is lowered, the carrier liquid in which the MNPs are dispersed, freezes (in the ZFC and FC measurements), the easy axis of the MNPs also “freezes” in random directions [22]. When this occurs, physical rotation of the MNPs (Brownian relaxation) is completely blocked, so that only the direction of the magnetic moment of the MNP can change (Néel relaxation). 

The zero-field cooled, and field cooled data acquired in this study indicate that stronger interactions led to a higher blocking temperature, as shown in Figure 3a–c. Interaction effects were also observed when the same nanoparticle systems were exposed to the AC magnetic field. Because of the rigid confinement of nanoparticles in the polystyrene matrix (see Figure 1b for Si/PS/Fe_3_O_4_), strong nanoparticle interactions limited Brownian relaxation, thus generating lower heat compared with other systems where nanoparticles are dispersed in the liquid. All data presented in this study suggest that dipole interactions play a critical role in modifying the relaxation mechanism. As shown in Figure 1b for Si/PS/Fe_3_O_4_, the densely packed Fe_3_O_4_ nanoparticles were confined in the spherical polystyrene matrices with much smaller inter-particle spacing. Therefore, the blocking temperature was directly affected by the inter-particle interactions.

In the model of Néel relaxation, the relaxation time is related to the particle’s individual magnetic anisotropy. In this study, the nanoparticle surfaces were modified for changing the energy barrier (ΔE) [23,24]. Thus, the magnetic anisotropy of the particles should be treated in terms of an effective magnetic anisotropy ($K_{eff}$). A phenomenological formula was proposed relating the effective magnetic anisotropy to particles the diameter (*D*), and is given as follows [23,25]:(2)$$K_{eff}=\frac{\Delta E}{V}=K_{V}+\frac{6K_{S}}{D}$$
where $K_{V}$ is the effective bulk anisotropy and $K_{S}$ is the effective surface anisotropy.

Figure 3a shows the ZFC and FC magnetization measurements at a field amplitude of 50 Oe for EMG 900 and EFH1, where Fe_3_O_4_ nanoparticles of 10 nm in diameter were dispersed in Isopar-M at a concentration of 48 mg/mL. The ZFC curve showed a broad peak at ~81 K for EMG 900 and at ~114 K for EFH1. Both samples exhibited a sharp cusp at approximately ~197 K (Figure 3a), which was associated with the freezing of Isopar-M carrier liquid at 196 K and the resultant blocking of movement of each MNP. The higher blocking temperature of EFH1 can be attributed to the influences of the larger distribution of particle sizes. The particle blocking and carrier liquid freezing effects in Fe_3_O_4_ nanoparticles were also reported in the study of Morales et al. [26] On the other hand, for the uniformly dispersed EMG 700 system, the ZFC and FC magnetization measured at the field amplitude of 50 Oe showed a broad peak at ~210 K, while that of the EMG 605 sample upwardly shifted the blocking temperature to ~240 K (Figure 3b). Both samples contained MNPs with the same average diameters of 10 nm, but the shift in blocking temperature of EMG 605 indicated a stronger dipolar interaction. The overlap of FC and ZFC at room temperature confirmed that the particles were indeed superparamagnetic. It should be noted that, even if all samples contained similar MNPs, the particle aggregation in the water-based sample led to an increased effective particle diameter, which will enhance inter-particle interaction, increase the blocking temperature, and reduce hysteresis loss. In contrast, the aggregation was much smaller in the Isopar-M-based fluid, resulting in a lower blocking temperature and higher hysteresis loss.

The effect of dipole interactions on the blocking temperature can also be investigated by changing the concentration or physical confinement of the MNPs. Vargas et al. [27] observed that for spherical magnetic nanoparticles with a narrow size distribution, by increasing the concentration, the inter-particle spacing was shorter, resulting in higher blocking temperatures. Similar behaviors were also reported in several other experimental and theoretical studies [28]. 

In this work, instead of concentration-dependent ZFC and FC measurements, we focused on a unique NP system (Si/PS/Fe_3_O_4_), where ~10 nm diameter Fe_3_O_4_ NPs were physically confined and densely packed in the spherical polystyrene matrices (Figure 1b). Figure 3c shows the ZFC and FC measurements of Si/PS/Fe_3_O_4_ at a field amplitude of 50 Oe, dispersed in water. It can be seen from Figure 3c that Si/PS/Fe_3_O_4_ exhibited a large upward shift in blocking temperature to 263 K. As we observed earlier, the blocking temperature of EMG 605 with more severe aggregation in the water was 30K higher than EMG 700 system with less aggregation. In the Si/PS/Fe_3_O_4_ sample, the MNPs were physically confined in the spherical matrices with much shorter inter-particle spacing, thus effectively increasing the dipolar interactions. Consequently, both EMG 605 and Si/PS/Fe_3_O_4_ exhibited an upward shift in blocking temperature, which is consistent with the previously reported studies [27,28].

The complex susceptibility $\left(\chi \left(w\right)=\chi^{\prime }\left(w\right)-i\chi^{\prime \prime }\left(w\right)\right)$ of the samples was measured over a wide frequency range using two complementary techniques: a slit toroid (100 Hz–10 MHz) [15] and a short-open coaxial line (SOCL) technique (10 MHz–7 GHz) [16]. The real and imaginary parts of the permeability were related to the susceptibility by ${µ}^{\prime }\left(w\right)=1+\chi^{\prime }\left(w\right)$ and ${µ}^{\prime \prime }\left(w\right)\approx \chi^{\prime \prime }\left(w\right).$ In the slit toroid inductively coupled transformer technique, a magnetic sample was placed in a gap that was machined through the cross section of a ferrite toroid. A swept frequency AC magnetic field was then inductively coupled to the toroid and the change in the toroid/sample inductance was measured with a lock-in amplifier. More details about the slit toroid technique can be found in [15]. In the SOCL technique, coaxial cells filled with liquid nanoparticle samples were terminated with either a short (s) or an open (o) endcap. The reflection coefficient $\Gamma_{s}$ and $\Gamma_{o}$ of the short and open circuit coaxial cavities was measured using a vector network analyzer (Agilent N5224A) and the complex permeability $\left(µ\left(w\right)={µ}^{\prime }\left(w\right)-i{µ}^{\prime \prime }\left(w\right)\right)$ of the samples is determined using the following formula [16]:(3)$$µ=\frac{1}{\beta_{o}l}\sqrt{\frac{\left(1+\Gamma_{s}\right)\left(1+\Gamma_{o}\right)}{\left(1-\Gamma_{s}\right)\left(1-\Gamma_{o}\right)}}\arctan\left(\sqrt{-\frac{\left(1+\Gamma_{s}\right)\left(1-\Gamma_{o}\right)}{\left(1-\Gamma_{s}\right)\left(1+\Gamma_{o}\right)}}\right)$$
where $\beta_{o}$ is the free space propagation constant and l is the sample depth.

The SOCL technique can be used up to a frequency where a standing wave resonance begins to dominate the reflection coefficient: this resonance occurs when the length of the sample region is $l=\frac{m\lambda }{4}$ (for short circuit configuration), where *m* is a natural number and λ is the wavelength in the sample. Therefore, to cover a wide frequency range, both the sample depth l and dielectric constant (ε) of the carrier liquid have to be sufficiently small. Thus, we only reported on the complex permeability of the EMG 900 and Si/PS/Fe_3_O_4_ samples suspended in Isopar-M (dielectric constant, ε = 2.05). The water in the Si/PS/Fe_3_O_4_ sample was removed by evaporation and Si/PS/Fe_3_O_4_ nanospheres were re-suspended in Isopar-M by sonication. 

Figure 4a,b shows the real part (imaginary part) of the susceptibility of the water-based EMG 700 sample at different concentrations, measured using the slit toroid technique. As shown in this figure, the real part of the susceptibility increased with increasing concentration, but monotonically decreased with the increasing frequency, while the imaginary part had a loss peak structure that appeared at frequency of 1 kHz–10 kHz. As shown in Figure 4b, as the concentration increased, the susceptibility loss progressively increased but the loss peak shifted to a lower frequency. It has been reported in several studies that the susceptibility loss peak is dominated by Brownian relaxation in the kHz range and by Néel relaxation in the ~MHz range [15,29]. For Brownian relaxation, the relaxation time is defined by $\tau_{B}=\frac{1}{f_{B}}=\frac{3\eta V_{H}}{k_{B}T}$, where η is the viscosity of carrier liquid (for water η = 0.99 mPa-s); $V_{H}=\frac{4\pi }{3}{\left(\frac{d_{h}}{2}\right)}^{3}$ is the hydrodynamic volume of the particles, and *T* is the absolute temperature. Using *d_h_* = 263 nm for the EMG700 sample, we found a value of $f_{B}$ to be 146 Hz. Because of the large particle size distribution, a broader peak appeared between the 1kHz and 100 kHz range for the EMG 700 sample, which can be attributed to Brownian relaxation. With increasing the concentration, the resonance peaks shifted toward the low frequency region, as shown in Figure 4b. 

The shift in resonance frequency with the increasing concentration could be described by the relaxation model presented by Feldman et al. [30]. As the particles are dispersed in a carrier fluid, surface charge starts to develop on the NP surfaces and eventually a stationary layer of fluid will accumulate. When the particles undergo Brownian relaxation in the presence of an AC magnetic field, the entire NPs physically rotate, carrying some bulk carrier fluid along. Usually, the distance from the stationary surface layer of the fluid to the bulk carrier fluid is much smaller compared with the particle−particle separation distance. Thus, the viscosity of the carrier fluid can be assumed to be constant with the increasing concentration. The inter-particle distance ($D_{p-p}$) decreased with concentration and obeyed the relationship $D_{p-p}=\frac{d_{h}}{2}{\left(\frac{4\pi }{3\varphi }\right)}^{1/3}$, where d_h_ is the hydrodynamic diameter and *φ* the volume fractions of NP. For 25 mg/mL and d_h_ = 40 nm, $D_{p-p}$ is ~190 nm. At a concentration of 278.4 mg/mL, this value was even reduced to 85 nm. Based on these calculations, the inter-particle distance was seen to progressively decrease with the increase in concentration. Meanwhile, the effective particle diameter increased due to fluid moving with particles, resulting in a longer relaxation time. The peak frequency of the relaxation time with the increasing concentration followed an exponential behavior, as observed in our study (Figure 4c). According to Feldman et al., for a unit macroscopic volume containing N dipoles at high concentrations (>5% volume), the dipole correlation function (DCF) is the sum of three exponential terms, representing intermolecular motion, Brownian motion, and slow motion [30]. A slow-motion contribution arises due to both the macromolecule motion and Brownian motion. At a lower concentration, Brownian motion has an isotropic relaxation time, but as the concentration becomes higher, the low frequency part of DCF becomes complex and can be expressed by the sum of two exponential functions, which correspond to anisotropic Brownian motion and slow motion [30]. It was shown in their study that the anisotropic Brownian motion has the same relaxation time, but the slow-motion relaxation time progressively increases due to the smaller inter-particle distances with increasing concentration. The complex susceptibility behavior of EFH1 dispersed in Isopar-M, as shown in Figure 4d,e, was different from the water-based EMG 700 ferrofluid. In the Isopar-M ferrofluid, which has higher viscosity than water, both the Néel and Brownian relaxation produced a much broader absorption peak than in the water-based ferrofluid. For EFH1/Isopar-M-based ferrofluid, as shown in Figure 4e, the imaginary susceptibility was negative at lower frequencies, which is attributed to the calibration error of the network analyzer for pure Isopar-M media. As no obvious peak was observed from the EFH1/Isopar-M-based sample in a low frequency region, Brownian relaxation was considered not the dominant mechanism in this sample due to the highly viscous media.

To check the validity of the dipole interaction model for Brownian relaxation, the complex susceptibility measurements were made on another water-based EMG 605 ferrofluid (Figure 4f). The nanoparticles of this ferrofluid were found to be severely aggregated, thus with strong dipole interactions. A large shift in Brownian relaxation was observed to the lower frequency for EMG 605 compared with the water-based EMG 700 ferrofluid. As discussed earlier, the aggregation in NP systems increased the effective particle diameter, resulting in a longer relaxation time, which was found to be consistent with the complex susceptibility measurements of the EMG 605 ferrofluid. An AC susceptibility measurement as a function of frequency on the confined MNP system (Si/PS/Fe_3_O_4_) was reported earlier in one of our previous articles [31]. A broad distribution of Brownian relaxation was observed for this sample, which peaked at ~300 Hz. The distribution of the relaxation frequency was found to be fitted quite well the logarithmic distribution of the particles’ hydrodynamics diameter (~70–200 nm), as measured by a dynamics light scattering technique for the Si/PS/Fe_3_O_4_ sample. 

The complex permeability of the Isopar-M-based ferrofluids were measured using the SOCL technique in order to cover as wide of a frequency range as possible (Figure 5a). Figure 5b,c shows the real and imaginary permeability of the EMG 900 sample. With the increasing frequency, the real part of susceptibility decreased and the imaginary part of the permeability increased, showing a peak at ~1.12 GHz. No concentration dependent shift in the resonance peak was observed in this study. 

The peak at ~1.3 GHz was assigned to gyromagnetic resonance absorption in previous studies on the spinel ferrite Fe_3_O_4_ structure [32,33]. Figure 5d shows the complex permeability of the Isopar-M-based Si/PS/Fe_3_O_4_ and EMG 900 samples at a concentration of 25 mg/mL. The microwave absorption peak was found to be shifted to ~2.92 GHz for Si/PS/Fe_3_O_4_ compared with the peak of ~1.12 GHz for the EMG 900 ferrrofluid. The difference in the absorption peaks between the two samples can be explained by anisotropy of these two systems. In Equation (2), we show effective anisotropy as the sum of bulk anisotropy and surface anisotropy (particle size dependence). As magnetocrystalline anisotropy arises due to the spin orbit interaction, it is assumed that the gyromagnetic resonances of these samples exhibit magnetocrystaline anisotropy at a GHz frequency. The gyromagnetic resonance frequency is related to magnetic anisotropy by the following relationship:$f_{r}=\frac{\gamma H_{K}}{2\pi }$, where γ = 28,024.95 MHz/T is the gyromagnetic ratio; $H_{K}=\frac{4\left|K_{1}\right|}{3{µ}_{0}M_{S}}$ being the internal field, where $K_{1}$ is the magnetocrystalline anisotropy in unit of J/m^3^ and $M_{S}$ is the saturation magnetization in A/m [6]. From the above equations, we can see that the resonance frequency would increase when increasing the anisotropy. Using the technique as shown in Section 3.4 of reference [2], saturation magnetizations of 290 kA/m and 309 kA/m were obtained for EMG900 and Si/PS/Fe_3_O_4_, respectively. For the resonance peak at ~1.12 GHz, the magnetocrystalline anisotropy was found to be 55 kJ/m^3^ for EMG 900, and 152 kJ/m^3^ for Si/PS/Fe_3_O_4_ at the peak at ~2.92 GHz. 

In the ZFC and FC measurements, the blocking temperature (proportional to the effective anisotropy constant) of Si/PS/Fe_3_O_4_ was found to be substantially higher than that for EMG 900, resulting in a larger upward shift in the gyromagnetic resonance frequency for Si/PS/Fe_3_O_4_. Using the core particle diameter of 10 nm and blocking temperature from the ZFC and FC measurements, the effective magnetic anisotropy constant of Si/PS/Fe_3_O_4_ and EMG 900 samples was found to be 176 kJ/m^3^ and 54 kJ/m^3^, respectively. Considering the densely packed nanoparticles inside the spherical polystyrene matrices with a close inter-particle distance, we believe that the effective anisotropy was high in Si/PS/Fe_3_O_4_ due to much stronger dipole interactions between the particles in the polystyrene matrix.

Time-dependent heating measurements were performed on two water-based samples, EMG 700 and EMG 605, using an Ameritherm induction heating system (Nova-3) at different field strengths and a fixed frequency of 335 kHz, as shown in Figure 5e. According to the literature, the specific absorption rate (SAR), which is the energy absorbed by the materials, is proportional to the initial heating rate of the ferrofluid, and the SAR should increase in line with the square of the magnetic field strength based on Néel and Brownian relaxation [34]. It should be noted that the initial heating rate was determined using the first 200 s of the plot shown in Figure 5f. By fitting the initial heating rate vs. magnetic field, we found that the measured initial heating rate deviated from the square dependence. This variation was greater for EMG 605 compared with EMG 700. From the complex susceptibility measurements, we observed a very small contribution of Brownian relaxation at 335 kHz for both samples; thus, most of the heating could be assumed to be coming from Néel relaxation and hysteresis loss. The TEM and DLS results show that EMG 605 had considerable agglomeration, leading to an increased effective particle size. In our previous work [2], we showed that the SAR of single domain particles progressively decreased when increasing the concentration, and chain formation was more favorable for a highly aggregated system. Therefore, hysteresis loss plays a critical role in generating heat for such nanoparticle systems. To perform a comparative study between the two samples under a magnetic field, we analyzed the hysteresis loss of these two samples under a DC magnetic field. DC magnetization measurements show a larger hysteresis loop of EMG 700 than that of EMG 605, as shown in the inset of Figure 2a. Under the same AC magnetic field for both samples (EMG 700 and EMG 605), we observed that the larger hysteresis loop of the former produced more hyperthermia heating than the latter. 

To investigate the effect of dipole interactions and the contribution carrier fluid on magnetic hyperthermia heating, time dependent heating measurements were performed on the three samples with particles of different configurations and were dispersed in various viscous media. A detailed study determining the SAR of the different nanoparticle systems from the initial heating has been thoroughly discussed in our previous article [2]. In this study, we measured the overall temperature changes of EMG 700, Si/PS/Fe_3_O_4_, and Isopar-M-based EMG 900 samples at a frequency of 335 KHz and field strength of 97 Oe and identified the dominant heating mechanism. As can be seen from Figure 5f, EMG 700 had a much greater temperature change (ΔT = 34.3 °C) than that of Si/PS/Fe_3_O_4_ (ΔT = 1.5 °C) after 10 min of exposure to the magnetic field. The significant difference in temperature change can be attributed to the reduced Brownian relaxation and stronger dipolar interactions in the Si/PS/Fe_3_O_4_ system. As shown in Figure 3, both zero field cooled and field cooled data indicate strong interactions in the Si/PS/Fe_3_O_4_ system, leading to a higher blocking temperature. The strong interactions that result from the rigid confinement of nanoparticles in the spherical polystyrene matrices (Figure 1b) have significantly reduced the Brownian relaxation effect (as particles are physically confined in the polystyrene matrices), thus generating less heat compared with other systems where the nanoparticles are dispersed in the liquid. As shown in Figure 1b, the dark spots are Fe_3_O_4_ nanoparticles with much smaller inter-particle spacing, thus with much stronger interactions [35,36,37,38]. These experimental results suggest that dipole interactions play a critical role in modifying the relaxation mechanism.

Furthermore, the EMG 900 sample with MNPs dispersed in more viscous liquid (Isopar-M) was found to exhibit less of a temperature change (ΔT =27.4 °C) compared with the EMG 700 sample, and the former even had a larger hysteresis loss. The nanoparticle dissipation of energy via hysteresis loss could be defined by the expression: $SAR={µ}_{o}Af$, where *A* is the hysteresis loop area and *f* is the frequency of the applied AC magnetic field. Under the same applied AC magnetic field, a relative comparison was made between the two samples in terms of the hysteresis loop area from the DC magnetization measurements. We obtained an SAR value of 31.41 Watt/mL for EMG 700 and 41.87 Watt/mL for EMG 900. Even though the SAR due to the hysteresis loss in EMG 900 was 34.6% higher than that of EMG 700, the temperature change of EMG 900 due to the magnetic heating was 20.1% lower than the EMG 700 at a frequency of 335 kHz. These experimental results suggest that, at 335 kHz, all three mechanisms of heating (Néel, Brownian, and Hysteresis) are effective in the EMG 700 and EMG 900 samples. The lower heating output of EMG 900 can be attributed to the reduced Brownian relaxation due to more viscous media. A similar conclusion was drawn by Vallejo-Fernandez et al., where they observed the reduction in heating of almost 43% by dispersing the sample in a medium of higher viscosity [34].

## 4. Conclusions

In conclusion, we have investigated the effects of magnetic dipole interactions on the dynamic loss mechanisms for medical therapeutics [4,5,6,7,8,9,10,11,36,37,38,39,40,41]. It should be noted that iron oxide nanoparticles have also been applied recently for solar harvesting and energy generation in multifunctional building skins [1,42,43]. The Fe_3_O_4_ nanoparticles can be structurally tailored for spectral selective solar harvesting and energy generation via localized plasmonic resonance [1]. Both DC and AC measurements are effective experimental approaches for studying the effects of the dipole interactions on blocking temperature and relaxation dynamics of different nanoparticle systems. These systems are characteristically different in terms of inter-particle interactions, as the nanoparticles are spatially configurated in various media, such as liquid or polymer nano spheres. While the nanoparticles have more freedom to move and rotate in liquid, they are rigidly confined in the polymer nano spherical matrices with much smaller inter-particle spacings, resulting in different magnetic behaviors.

We measured the AC responses of different samples, and determined the magnetic anisotropy based on the resonance of each sample. This allowed us to quantitatively analyze the effect of the particle interaction on anisotropy. We investigated the effects of magnetic dipole interactions on the dynamic loss mechanisms via both DC and AC measurements of five Fe_3_O_4_ systems with different surfactants, carrier liquids, and nanoparticle confinement. The DC magnetization measurements showed a direct correlation between the hysteresis loop and hyperthermia heating via dipole interactions under AC field. The blocking temperature was found to be substantially higher in the systems with strong dipole interactions. Complex susceptibility measurements using slit the toroid technique showed a pronounced peak over the frequency range of 1kHz–10kHz for EMG 700, attributable to Brownian relaxation. By increasing the magnetic concentration, the Brownian relaxation peak shifted to a lower frequency. Complex permeability measurements using the SOCL technique showed a gyromagnetic resonance of Fe_3_O_4_ nanoparticles at ~1.12 GHz. Si/PS/Fe_3_O_4_ and EMG 900 exhibited different resonance frequencies in the microwave region. These changes in blocking temperature and resonance dynamics suggest the spin dynamics correlation with the particles’ effective magnetic anisotropy, which can be altered by nanoparticle surface modifications and physical confinement. 

## Figures and Tables

**Figure 1 materials-16-00496-f001:**
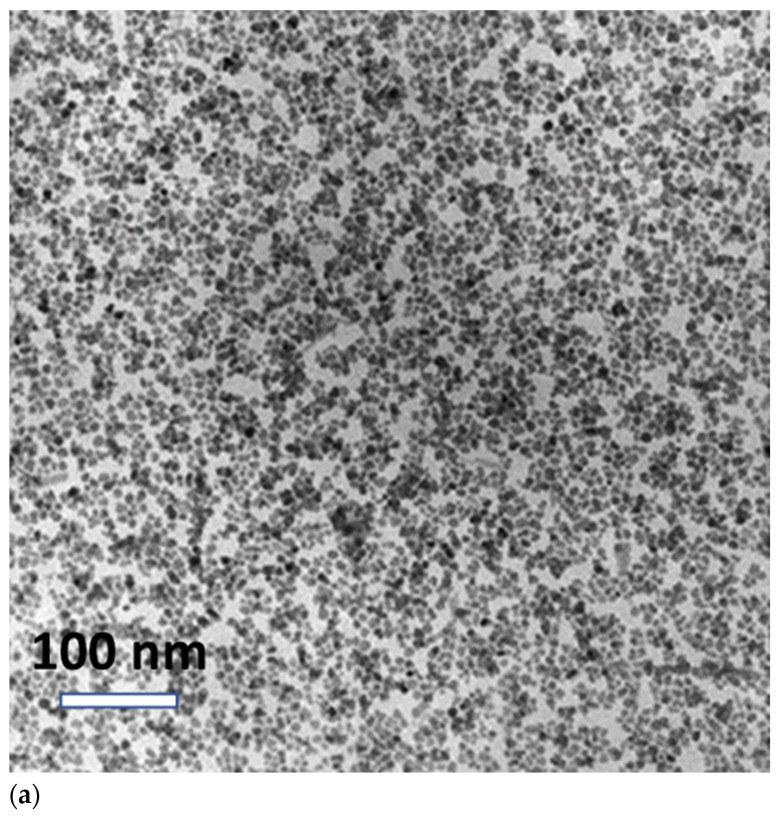
(**a**) Transmission electron microscopy (TEM) image of the commercial samples (image supplied by Ferrotec, Bedford, NH, USA). (**b**) Transmission electron microscopy image of Si/ PS/Fe_3_O_4_. The nanoparticles of Fe_3_O_4_ (darker spots) are embedded in the spherical polystyrene matrices (lighter spheres).

**Figure 2 materials-16-00496-f002:**
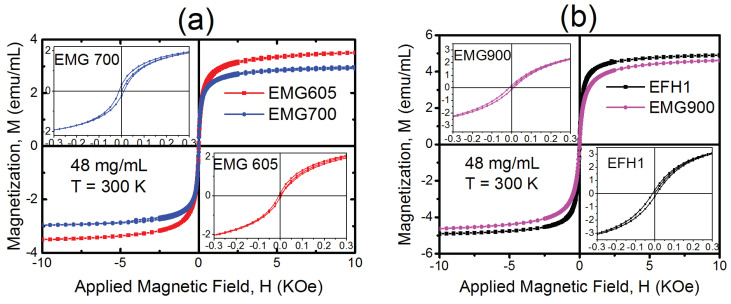
DC magnetization behavior of (**a**) water-based EMG 605 and EMG 700 samples, and (**b**) isopar −M −based EMG 900 and EFH1 samples, the inset shows the corresponding low field portion of the data.

**Figure 3 materials-16-00496-f003:**
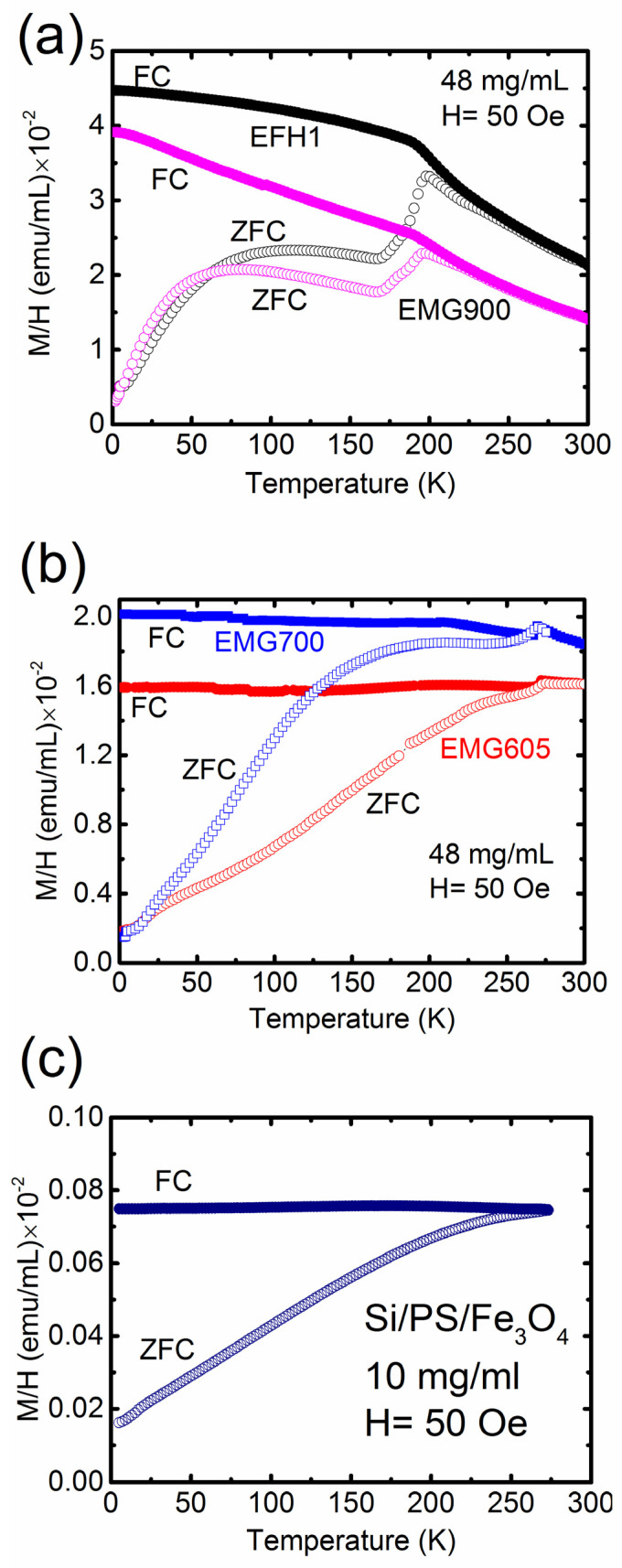
ZFC and FC magnetization behavior of the (**a**) Isopar-M-based EMG 900 and EFH1 samples; (**b**) water-based EMG 700 and EMG 605 samples; (**c**) water-based Si/PS/Fe_3_O_4_ sample.

**Figure 4 materials-16-00496-f004:**
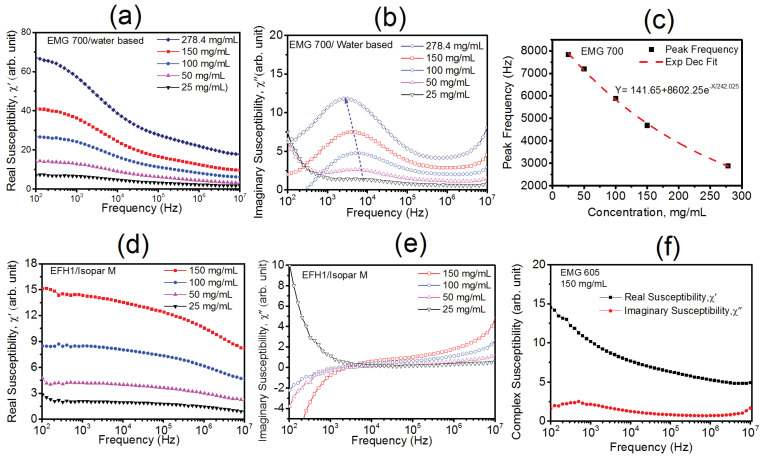
(**a**,**b**) Real (**a**) and imaginary part (**b**) of susceptibility of the water-based EMG 700 samples at different concentrations. (**c**) Fitting of the peak frequency of Brownian relaxation vs. concentration for the EMG 700 sample. (**d**,**e**) Real (**d**) and imaginary (**e**) part of susceptibility of the Isopar-M-based EFH1 sample at different concentrations and (**f**) complex susceptibility of the EMG 605 sample at a concentration of 150 mg/mL.

**Figure 5 materials-16-00496-f005:**
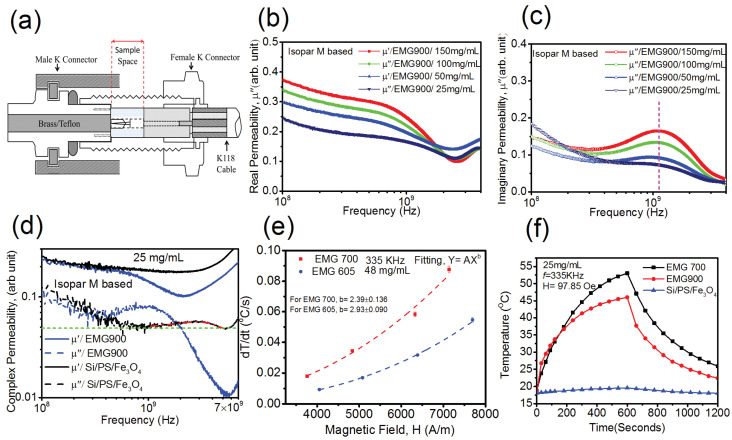
(**a**) K connector (Anritsu) cavity (sample depth= 2.1 mm) for the measurement of the complex dielectric properties of liquid ferrofluid. The short circuit cavity was designed by placing a Brass rod into the inner diameter of male K connector (shown as grey in the image) for open circuit cavity brass rod, and the pin of the male connector was replaced by a similar diameter Teflon rod and pin(**b**,**c**). Real (**b**) and imaginary (**c**) part of the permeability of the Isopar-M-based EMG 900 samples at different concentrations. (**d**) Complex permeability of Isopar-M-based Si/PS/Fe_3_O_4_ and EMG 900 samples. (**e**) Initial heating rate vs. magnetic field curve, measured at a frequency of 335 KHz at a sample concentration of 48 mg/mL for water-based EMG 700 and EMG 605 samples, and (**f**) time dependent temperature curve (heating first 10 min, cooling next 10 min) of water-based EMG 700 and Si/PS/Fe_3_O_4_ and Isopar-M-based EMG 900 samples.

## Data Availability

The data that support the findings of this study are available from the corresponding author upon reasonable request.

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
