# Peer review of "Effect of Dipole Interactions on Blocking Temperature and Relaxation Dynamics of Superparamagnetic Iron-Oxide (Fe3O4) Nanoparticle Systems"

_materials, 2023, doi:10.3390/ma16020496_

Round 1
Reviewer 1 Report
The authors have investigated the effects of dipole interactions on the dynamic loss mechanisms via DC and AC measurements of five Fe3O4 systems with different surfactant, carrier liquid, and nanoparticle confinement.
The article is well organized and contained a new information which will be interesting for the readers of Materials journal.
The experimental data were obtained using the appropriate modern methods/
More details about instruments (TEM, etc.) must be presented with the indication of the setups'models, manufacture with the country and city information.
Author Response
Reviewer #1: More details about instruments (TEM, etc.) must be presented with the indication of the setups' models, manufacture with the country and city information.
Response: We have added the required information and highlighted it in red in the revised manuscript.
“The TEM images were taken using a JEOL 2010F for characterization of the nanoparticles. Samples were developed by placing a drop of the MNP solution on a carbon coated copper grid and letting it dry at room temperature. The mean hydrodynamic diameter and size distribution of the MNPs dispersed in water were measured by Zetasizer (Nano Series, Malvern Instruments).”
Reviewer 2 Report
Dear authors,
The experimental results of this study is very good. The data is complete and well presented. There are several suggestions need to be addressed.
1. Please double check a few misspellings.
2. Line 93. It is better if the authors mention the whole procedure for producing the specimens.
3. The characteristics of each samples need to be mentioned
4. In the conclusion, please mention the suitability of the experimental results for the purpose of this study
Author Response
Reviewer #2:
- Please double check a few misspellings.
Response: Spelling check has been completed
- It is better if the authors mention the whole procedure for producing the specimens.
Although the experimental procedures were published previously in detail (ref. 2, 3), we added some brief descriptions in the text and highlighted them in red.
“Briefly, a chemical coprecipitation method3 was employed to synthesize the Fe3O4 nanoparticles with an average particle size ∼10 nm. The nanoparticles were dispersed in situ in octane to form a ferrofluid. A miniemulsion was developed by adding the Fe3O4 ferrofluid to the aqueous solution with sodium dodecyl sulfate (SDS) as surfactant. This miniemulsion containing Fe3O4 aggregations was mixed with another miniemulsion made of the styrene monomer droplets. The mixture of both miniemulsion was reacted at 80 °C for 20 h to form the polystyrene nanospheres with Fe3O4 nanoparticles embedded in the spherical matrices. The nanosphere surfaces were functionalized with silica by following a previously reported process3 to complete the synthesis of Si/Polystyrene/Fe3O4.”
- The characteristics of each samples need to be mentioned
Response: We have added further descriptions of the characteristics of each sample in terms of morphology and interactions and highlighted them on p. 5 in the revised version.
“By comparing Figure 1a and 1b, one can see the distinctive characteristics of the Fe3O4 nanoparticles and the Si/PS/Fe3O4 nano spheres. For Si/PS/Fe3O4, the Fe3O4 nanoparticles are physically confined in the polystyrene spherical matrices. In this situation, the interparticle separation is small which is on the order of one particle diameter, resulting in strong dipole interactions. However, in ferrofluid, the Fe3O4 nanoparticles are free to move in the solution with much weaker inter-particle interactions. These two unique behaviors will result in significant differences in magnetic dipole interactions.”
- In conclusion, please mention the suitability of the experimental results for the purpose of this study.
Response: We have added more descriptions in the conclusion on the experimental approaches being suitable for characterization of the effects of dipole interactions and highlighted them in red in the revised version.
“In conclusion, we have investigated the effects of magnetic dipole interactions on the dynamic loss mechanisms. Both DC and AC measurements are the most effective experimental approaches for studying the effects of dipole interactions on blocking temperature and relaxation dynamics of different nanoparticle Systems. These systems are characteristically different in terms of interparticle interactions since the interparticle spacings are set differently. On the other hand, they are also dispersed in different media, liquid, or rigid polymer; while the particles have more freedom to move in the former, the latter confines them in the spherical matrices. Only in this fashion, one can correlate the interparticle interactions and related magnetic behaviors. None of this has been reported on the drastically different nanoparticle systems.”
Reviewer 3 Report
1. Please comment on the use of oil and water solvents
2. Can the author tell which influence affects blocking temperature
Author Response
Reviewer 3
- Please comment on the use of oil and water solvents
Responses: In this study, we used both commercial oil-based (EMG900) and water-based (EMG 700) Fe3O4 MNPs (~ 10 nm diameter). We found no significant differences between them in terms of the magnetic behaviors since both ferrofluids contain nanoparticles that are free to move/rotate in the liquid, while the Fe3O4 nanoparticles are rigidly confined in the matrices of the polystyrene in the Si/PS/Fe3O4 sample. So, the major differences are found between the Si/PS/Fe3O4 and EMG samples.
- Can the author tell which influence affects blocking temperature
Response: As shown in the TEM image of Si/PS/Fe3O4 in Figure 1b, the densely packed Fe3O4 nanoparticles are confined in the spherical polystyrene matrices. The dark spots are Fe3O4 nanoparticles with much smaller interparticle spacing, therefore, it is the very strong interactions that affect blocking temperature. We added a paragraph on this issue in the Discussion on page 8.
“Zero field cooled and Field cooled data acquired in this study indicate that stronger interactions lead to higher blocking temperature as shown in Figure 3 (a-c). Interaction effects were also observed when the same nanoparticle systems were exposed to AC magnetic field. Due to rigid confinement of nanoparticles in the polystyrene matrix [see Figure 5(f) for Si/PS/Fe3O4], strong nanoparticle interactions limited Brownian relaxation, therefore generating lower heat compared to other systems where the nanoparticles are dispersed in the liquid. All data presented in this study suggest that dipole interactions play a critical role in modifying the relaxation mechanism. As shown in Figure 1b for Si/PS/Fe3O4, the densely packed Fe3O4 nanoparticles are confined in the spherical polystyrene matrices with much smaller interparticle spacing. Therefore, the blocking temperature is directly affected by the interparticle interactions.”